# Effects of Radiation on Cross-Linking Reaction, Microstructure, and Microbiological Properties of Whey Protein-Based Tissue Adhesive Development

**DOI:** 10.3390/polym14183805

**Published:** 2022-09-12

**Authors:** Ning Liu, Guorong Wang, Mingruo Guo

**Affiliations:** 1Department of Nutrition and Foods Sciences, The University of Vermont, Burlington, VT 05405, USA; 2College of Veterinary Medicine, Northeast Agricultural University, Harbin 150006, China

**Keywords:** whey protein isolate, gamma radiation, sterilize, tissue adhesive, cross-linking

## Abstract

Whey proteins are mainly a group of small globular proteins. Their structures can be modified by physical, chemical, and other means to improve their functionality. The objectives of this study are to investigate the effect of radiation on protein–protein interaction, microstructure, and microbiological properties of whey protein–water solutions for a novel biomaterial tissue adhesive. Whey protein isolate solutions (10%, 27%, 30%, 33%, and 36% protein) were treated by different intensities (10–35 kGy) of gamma radiation. The protein solutions were analyzed for viscosity, turbidity, soluble nitrogen, total plate count, and yeast and mold counts. The interactions between whey proteins were also analyzed by sodium dodecyl sulfate polyacrylamide gel electrophoresis and scanning electron microscopy. The viscosity of protein solution (27%, *w*/*w*) was increased by the treatment of gamma radiation and by the storage at 23 °C. The 35 kGy intensity irradiated soluble nitrogen (10%, *w*/*w*) was reduced to about half of the sample treated by 0 kGy gamma radiation. The effects of gamma radiation and storage time can significantly increase the viscosity of whey protein solutions (*p* < 0.05). Radiation treatment had significant impact on soluble nitrogen of whey protein solutions (*p* < 0.05). SDS-PAGE results show that the extent of oligomerization of whey protein isolate solutions are increased by the enhancement in gamma radiation intensity. Photographs of SEM also indicate that protein–protein interactions are induced by gamma radiation in the model system. Consistent with above results, the bonding strength increases by the addition of extent of gamma radiation and the concentration of glutaraldehyde. Our results revealed that the combination of gamma-irradiated whey protein isolate solutions and glutaraldehyde can be used as a novel biomaterial tissue adhesive.

## 1. Introduction

Sealing wounds and stopping hemorrhages on the battleground have been going on for thousands of years. From then on, the suture was created and has become a common way to treat wounds in modern medical treatment. Currently, various types of absorbable and non-absorbable sutures with different-size needles for biomaterials and tissue engineering techniques have shown a promising efficacy in clinical practice [1]. However, due to the difficulty of several manifest disadvantages establishes into dealing with wounds by suture, such as high infection rate, physical pain, and long healing time, there still remains a challenge to provide a satisfactory cure in tissue surface regeneration [1]. While mechanical suture techniques are a crucial technique for surgeons, suturing is usually imperfect for both the clinician and the patient [2]. Sharp injuries are considered to be the most pertinent concern due to the economic and psychological burdens, as well as the risk of communicable disease transmission [2]. Therefore, alternative methods to suture have been evaluated, including adhesive tapes, gums, or tissue adhesives.

Although great energy and financial resources have been invested in the research of tissue adhesive, recent developments in tissue adhesive strategies and techniques are limited in bonding biological epidermis. Tissue adhesives are usually classified as biological, composite biological, and synthetic [3]. Existing traditional adhesives, including cyanoacrylate glues or fibrin sealants are far from ideal, presenting with poor bonding strength in wet conditions, cytotoxicity, and sequela [4]. Moreover, some materials require additional surgical burden such as suturing or gluing with fibrin [5]. Even after covering the wounds physically with dressing materials, severe inflammatory responses may occur due to external stimuli and acidic degradation products (pH 2–3) [6], which prolong the healing process [5]. Even the whey protein has potential to be used as a functional polymer component for adhesives, the use of proteins as a component in tissue adhesives has been rarely studied.

Whey, the byproduct of cheese-making, is not fully used by human and animals every year. Preventing pollution caused by disposal of whey and recovering its nutrient value have been two major factors of interest of using whey for decades. The main utilizations of whey are for animal feed, fertilizers, or as an ingredient for human foods due to its high nutritional value, low price, and various of functions [7], as well as its excellent emulsification properties, low biological toxicity, and high biocompatibility [8]. These characteristics of this multi-functional biomaterial were focused on by recent advances in tissue engineering [9]. Whey protein isolate (WPI, >90% protein) is mainly composed of β-lactoglobulin (β-LG), α-lactalbumin (α-LA), immunoglobulin (IG), and bovine serum protein (BSA) [10,11]. WPI can form gelation induced by energy input due to denaturation, unfolding of protein chains, and aggregation of proteins into a 3-dimensional network by changing the hydrophobic interactions, thiol-disulphide bond exchange, and hydrogen bonding [11]. However, protein-stabilized emulsions are sensitive to temperature, ionic strength, and pH [12]. Consequently, sterilization strategies are urgently needed to improve the physical and chemical stability of WPI emulsions [13].

Non-thermal processing technologies [14], such as gamma radiation, have paid close attention and enormous interest in food industry [15]. The reason why gamma radiation has attracted so much attention is also because of its multiple effects of sterilization, bacteriostasis, and changing protein spatial conformation through energy transfer. It occurs owing to the formation of cross-links between protein chains via the disulfide bridge mechanism [16]. At present, gamma radiation has been used in the sterilization of food products because of its effectiveness and safety. Gamma radiation is able to penetrate the products and purge bacteria completely [17]. Moreover, aqueous solutions first exposed to gamma radiation generate hydroxyl radicals and hydrated electrons, which are capable in turn react with molecules to form covalent bonds [18]. Gamma radiation is employed to enhance water vapor permeability and chemical stability of protein-based films [19], mechanical resistance of milk protein gels [16], viscosity of whey dispersions [20], oil absorption, and emulsion capacities of cowpea protein isolates [15,21]. The benefits of food irradiation include disinfection, inhibiting sprouting, and pathogen killing to prolonging the shelf life of foods and delays ripening. Furthermore, gamma-irradiation could be used in pharmaceutical or biomedical applications.

The objectives of this study are to apply gamma radiation for sterilizing whey protein solution and to formulate a tissue adhesive using whey protein as the major polymer. The ideal tissue adhesive would have the following capabilities which are safe, biodegradable, effective, easily usable, affordable, and approvable by government regulatory agencies for use. Our results also provide a theoretical basis and feasibility study for clinical medicine to use the gamma radiation-sterilized WPI and a cross-linker, glutaraldehyde (GTA), as the ingredients of a tissue adhesive of a novel biomaterial.

## 2. Materials and Methods

### 2.1. Reagents

WPI with a protein content of 90% was purchased from Fonterra^TM^ (Auckland, New Zealand). GTA (50%) was purchased from Fisher Scientific Inc. (Pittsburgh, PA, USA). Aerobe count and yeast and molds count Petrifilms were purchased from the 3M^TM^ Petrifilm^TM^. Porcine skins were purchased from a local supermarket. Pre-filled buffer (99 mL) was purchased from Fisher Scientific Inc. (Pittsburgh, PA, USA).

### 2.2. Model System Preparation

Protein tissue adhesive was comprised of WPI solution and GTA solution. WPI powder was dissolved in deionized water at concentrations of 10%, 27%, 30%, 33%, and 36% (*w*/*w*). The solutions were mixed with a blender for 15 min at 1000 RPM. WPI solutions were stored in a refrigerator at 4 °C overnight. Defoamed WPI solutions were poured into 10 mL centrifuge tubes and labeled for protein concentrations and irradiated dosages. Various concentrations of WPI solutions were treated by different dosages (10–25 kGy) of gamma radiation (^60^Cobalt irradiator) at ambient temperature with presence of oxygen. The Gamma radiation was operated on a 220 V, 60 Hz, and 15 A. For soluble nitrogen detection and electron microscopic observation, 10% and 27% WPI solutions were treated by 10, 15, 20, 25, 30, 35 kGy of gamma radiation. The radiated WPI solutions were stored at room temperature. GTA (50%) was diluted with sterilized deionized water to 8.0% and 10.0% (*v*/*v*).

### 2.3. Microbial Analysis

Each sample (1 mL), respectively, was diluted to 10^−1^ with 9 mL sterile buffer and another 1 mL of the sample was diluted to 10^−2^ with 99 mL sterile buffer. 10^−3^ dilution was prepared by adding 1 mL 10^−1^ dilution to 99 mL of sterile buffer. Each different dilution was applied on aerobe count and yeast and molds count Petrifilms according to the manufacturer’s instructions. Triplicates for each sample were tested for aerobe, yeast, and molds. The top of films was lifted to expose the surface and 1 mL of each dilution (10^−1^, 10^−2^, and 10^−3^) from each sample was added inside a circular foam barrier on aerobe count Petrifilms and yeast and molds count Petrifilms. The Petrifilm has a circular bottom with a barrier and a top to enclose the sample within the Petrifilm. Petrifilms were gradually rolled down and distributed evenly by a plastic spreader and gas bubbles were excluded by this process. WPI solutions prior to gamma radiation treatment were also tested by aerobe count and yeast and molds Petrifilms. After a few minutes of drying, aerobe count Petrifilms were put into an incubator at 37 °C for 48 h and yeast and molds count Petrifilms were put at room temperature for at least 48 h. The readable plate count for Petrifilm is between 25 and 250 colony-forming units (CFUs). There will be splotches or spots surrounded by bubbles for molds if there are CFUs present.

### 2.4. Viscosity

Apparent viscosity of various concentrations of WPI solutions (27–36% protein) before and after irradiation at various dosages (0–25 kGy) was determined at room temperature (23 °C) by a Viscometer (Model DV-I Prime, Brookfield Engineering Labs Inc., Stoughton, MA, USA). Samples were tested with No. 3 spindle at 100 RMP. The viscosity reading on the screen was recorded after the spindle has been activated for 20 s. Three replicates were performed for each sample.

### 2.5. Gelation during Storage

Sample gelation was observed every other week during storage. If WPI solutions showed solidifying properties, then they are not suitable for the development of tissue adhesives.

### 2.6. Soluble Nitrogen

The pH 4.6-soluble nitrogen of WPI solutions was tested by the method of International Dairy Federation (IDF 1964a). A total of 4 mL of each sample combination was weighed, respectively, and added into 10 mL centrifuge tubes. About 4 mL of distilled water was transferred into each tube. Acetic acid (10%, *v*/*v*, 0.4 mL) was added into each tube, mixed, and kept for 10 min. Na acetate (1M, 0.4 mL) was added into each tube and mixed gently. Volume was made up to 10 mL with deionized water in all tubes. All samples of mixture were filtrated through Fisher Q 8 filter paper. The precipitate was removed and a sample (2 mL) of filtrate was taken for nitrogen determination by the Kjeldahl method.

### 2.7. Lap-Shear Bonding Strength

Lap-shear bonding strength was measured with the method of ASTM standard (ASTM 2005). Fresh porcine skin was cut into 5.08 cm × 2.54 cm pieces with a No. 15 Uniblade^TM^ surgical scalpel (AD Surgical, Sunnyvale, CA, USA). Two skin strips were adhered on two aluminum blocks with dermal side up with Gorilla^®^ super glue (The Gorilla Glue Company, Cincinnati, OH, USA). The test specimens were kept moist by wrapping with gauzes soaked with phosphate buffered saline (PBS) (Fisher Scientific, Fair Lawn, NJ, USA) and placed in an environmental chamber at 25 °C. About 80 μL of WPI solution and 20 μL of GTA solution were applied on the skin dermal side and mixed with a small steel spatula and then the two porcine skin strips were overlapped together by the fixture. The bonding area was 2.54 cm × 1.0 cm. Glued specimens were kept for 30 min and tested by an Instron 5566 machine (Instron Corporation, Norwood, MA, USA). Specimens were put tightly in the loading cell of the Instron and the test was started until the two porcine skin stripes separated. The maximum load (N) was recorded and the bonding strength (kPa) was calculated.

### 2.8. Sodium Dodecyl Sulfate-Polyacrylamide Gel Electrophoresis (SDS-PAGE)

SDS-PAGE was conducted with separating and stacking gels. The glass-plate sandwiches were assembled and sealed with Vaseline. Total volume of 80 mL sample buffer, separating gel, and stacking gel were prepared. About 50 μL sample was added into 10 mL sample buffer and mixed well. Equal amounts of samples were loaded on each lane for comparison. SDS-PAGE was performed at a constant current of 150 V for 10 min at 60 mA and then at 200 V mA at 25 mA for an additional 30–40 min at room temperature. Gels were fixed with 25% (*v*/*v*) propan-2-ol and 10% glacial acetic acid overnight and then stained with Coomassie Brilliant Blue Dye (Thermo Fisher Scientific, Waltham, MA, USA) at room temperature for 5 h. Then the gels were disdained with distained buffer for 2–3 h until the blue color was washed off to make the gels transparent and the bands clearly visible.

### 2.9. Scanning Electron Microscopy (SEM)

On two separate occasions, SEM was used for analyzing the microstructure of WPI solutions. For the first trial, whey protein adhesives were prepared by loading in an applicator to mix and spread the components. The material was extruded out onto a piece of Parafilm and cured at room temperature until the samples were dehydrated completely. Then the material was frozen in liquid nitrogen and fractured. The pieces were mounted on aluminum stubs with silver conductive paste and carbon coated and then sputter coated with Au/Pd (approximately 4 nm) (Hendricks 1983). Treated samples before and after irradiation were then scanned on a FEI Quanta 200 FEG ESEM Mark II scanning electron microscope (UMASS, Worcester, MA, USA) at an accelerating electron voltage of 10 kV. Second trial, SEM images were obtained at the University of Vermont. All images were recorded digitally using Scandium software 5.2 (Olympus Corporate, Germany).

### 2.10. Molecular Docking of WPI and GTA

The crystal structures of target proteins (α-lactalbumin and β-lactoglobulin) come from protein database (https://www.rcsb.org/ (accessed on 17 January 2022)). After using PyMol 2.1 software (Schrödinger Inc., New York, NY, USA) to delete irrelevant small molecules in protein molecules, the protein molecules were imported into Autodock Tools 1.5.6 software (Scripps Research Institute, La jolla, CA, USA) and then deleted water molecules, added hydrogen atoms, set atom types, and finally saved them as pdbqt files. The structure of GTA compound was from PubChem database (https://pubchem.ncbi.nlm.nih.gov/ (accessed on 17 January 2022)). The GTA molecule was imported into Autodock Tools 1.5.6 software and then added atomic charges, assigned atomic types, made all flexible keys rotatable by default, and finally saved them as pdbqt files.

The treated compounds were used as small molecule ligands and five protein targets were used as receptors. According to the interaction between small molecules and targets, the central position of grid box (alpha La: x_center = 18.26, y_center = 5.25, z_center = 6.19; beta LG: x_center = −0.26, y_center = 16.25, z_center = −7.19) and length, width, and height were set to 40 × 40 × 40. Finally, molecular docking and result analysis are carried out through Autodock Tool. PyMOL software 2.1 was used to visualize the binding results between compounds and proteins. In the calculation process, the Lamarckian genetic algorithm is used for molecular docking calculation. The independent docking operation is 100 times and the final docking structure is evaluated according to the combined free energy.

### 2.11. Statistical Analysis

SPSS^®^ Version 21 software for Linux (SPSS Inc., Chicago, IL, USA) was used for statistical analysis. The data were analyzed by ANOVA which analyzes the effects of gamma radiation treatment and storage time. Significant differences between group means of the variations were observed in partitioned components. The *p* < 0.05 is considered as significant differences.

## 3. Results

### 3.1. Effect of Radiation on Microbial Properties

The WPI solutions were tested for aerobic microorganisms and yeast and molds by 3M^TM^ Aerobic Count Petrifilm^TM^ and Yeast and Molds Petrifilm^TM^, respectively, for six months after gamma radiation treatment (10–25 kGy). No microorganism has been found during storage for treated samples in both aerobic count Petrifilms and yeast and molds Petrifilms (Table 1). The smallest dosage (10 kGy) of gamma radiation applied in this study can sufficiently sterilize WPI solutions and eliminate the presence of viable microorganisms. The tests for aerobic bacteria and yeast and molds before the treatment of gamma radiation were also conducted in Chinese Academy of Agricultural Sciences, Beijing, China. The colonies on aerobic count Petrifilms were 107, 43, and 93 for 10^−1^ dilution, so the mean of results was 810 CFU/mL. The mean of results was 200 CFU/mL on the yeast and molds Petrifilms. Bacteria that has been gamma irradiated usually frequently maintained metabolic activity and morphological integrity but it will lose the ability to multiply [22].

### 3.2. Viscosity

The viscosity was measured to explore the effects of gamma radiation on cross-linking reaction of WPI solutions. The viscosity of WPI solutions was influenced by the treatment of various dosages of gamma radiation. The viscosity before and after gamma radiation is shown in Figure 1. Comparing to WPI solutions treated by 0 kGy gamma radiation, the viscosities of protein solutions (33% and 36% protein) were significantly increased over time treated by 10 kGy gamma radiation, respectively (Figure 1A) (*** *p* < 0.01). However, the viscosities of protein solutions (27% and 30%) were not increased over time treated by 10 kGy gamma radiation. This may be due to the lower amount of protein that can cross-link in the solution at low concentrations. Figure 1B-E shows the changes in the visibility of protein concentrations at 27%, 30%, 33%, and 36% within six months after gamma radiation ranging from 0 kGy to 25 kGy. The viscosity of protein significantly increased with the enhancement of protein concentration, irradiation dosage, or storage time. Gamma radiation has the effects of fragmentation and aggregation. Proteins can be converted to higher molecular weight aggregates, due to inter-protein cross-linking reactions, hydrophobic and electrostatic interactions, as well as the formation of disulfide bonds [23]. Gamma radiation which can be assimilated by WPI solutions to produce hydroxyl radical and superoxide anion provided energy to induce physical and chemical conversions including aggregating and cross-linking proteins [24]. The denatured or aggregated WPC (whey protein concentrate) and WPI decreased symmetric shape and enhanced in volume fraction than the native molecules [25]. The sudden decline of viscosity in the data may be because of protein gelation due to gamma radiation and may also because of the interactions between proteins enhanced by gamma irradiation.

Viscosity is a measurement of pressure which comes from interactions or collisions of particles moving in different directions and velocities in fluid. Technically, with regard to homogeneous solution, the higher the concentration of the solution, the higher viscosity the fluid shows. In some reports, pH, temperature during gamma radiation, and hydration state also need to be taken into consideration [21]. In this study, the variables that change the viscosity include irradiation intensity, protein concentration, and storage time. In this study, temperature was kept constant at room temperature to eliminate interference. Gamma radiation may induce protein–protein interactions resulting in increasing of viscosity and bonding strength, decreasing of pH 4.6 soluble nitrogen and turbidity.

### 3.3. Gelation

The gelation of WPI solutions were observed every week after gamma radiation (Table 2). As long as WPI solutions form into gelatinous contents, they are not suitable for future experiments. Protein solutions (27%) treated by 25 kGy gelled within six-month storage. Protein solutions (30% and 33%) treated by 15, 17.5, 20, 25 kGy all formed into gelatinous contents during storage. Protein solutions at 36% treated by 15, 17.5, 20, 25 kGy gelled in first 3 months and treated by 12.5 kGy formed into gels in fifth month. Samples treated by 10 kGy show stability of gelation within six months. Gelation velocity of WPI solutions is proportional to irradiation intensity and protein concentration. The reason for the formation of gelation may be related to the irradiated intensity of gamma radiation, which increased the cohesive strength of adhesive materials [26]. The higher irradiation intensity and protein concentration also mean the higher cohesive strength in WPI solutions. High dose irradiation intensity can accelerate protein–protein cross-linking, which forms high polymers between proteins. These results inspired us to precisely choose the irradiation intensity and protein concentration to ensure that the best bonding effect can be achieved when applied.

### 3.4. Soluble Nitrogen

Gamma radiation affects functional and structural properties of proteins, such as nitrogen solubility. Changes in soluble nitrogen are affected by a number of different indices including protein denaturation, conformation, structure, and sequence of the protein molecules [15]. Soluble nitrogen also measures the effects of gamma radiation on cross-linking reactions of WPI solutions. In this study, the nitrogen solubility was decreased by gamma radiation due to protein-based conformational modification, shown in Figure 2A. The soluble nitrogen (10%, *w*/*w*) was decreased from 100% for untreated samples to 54.7% for samples treated by 35 kGy gamma radiation. The statistical analysis of soluble protein indicated that radiation treatment had significant impact on soluble nitrogen of whey protein solutions (*p* < 0.05). Our result indicated that soluble nitrogen of WPI solutions was negatively correlated to irradiation intensity, which may be because of the aggregation and cross-linking reaction of WPI after irradiation.

### 3.5. Turbidity

Turbidity is also an important criterion to judge the effect of gamma radiation on protein aggregation. We also continuously monitored the turbidity of protein after gamma radiation for four months (Figure 2B–D). The turbidity of protein (10% *w*/*w*) increased with the increase of gamma radiation intensity, but the turbidity of protein decreased significantly after long-term storage. After long-term storage, the turbidity of all proteins with the same concentration and irradiation intensity significantly decreased. This result is consistent with that of soluble nitrogen, indicating that the reduction of soluble nitrogen in WPI solutions after long-term storage of irradiation is the main reason to result in the decrease of turbidity.

Cross-linking reaction caused by gamma radiation was also determined by SDS-PAGE. On the base of electrophoretic mobility, proteins are conformed to several lines on the gel depending on their molecular weight. SDS is used to form linear-structural proteins and cause a negative charge so that proteins migrate from negative electrode to positive electrode. Protein profiles of WPI bands of β-LG and α-LA under different dosages of gamma radiation are shown in Figure 2E. The SDS-PAGE profiles of WPI solutions (10% and 30%) both show that proteins were cross-linked in the samples treated by gamma radiation loading equal volume of samples. The protein profiles indicate that proteins were aggregated to larger molecular weights than samples untreated by gamma radiation. Protein bands irradiated by gamma rays are less bright and smaller than unirrigated protein bands. Moreover, this trend is also more obvious with the increase of irradiance intensity. Any amino acid radical that is formed within a peptide chain could cross-link with an amino acid radical in another protein [27]. Other reports have shown similar findings that lately created myosin bands after gamma radiation possess higher molecular weight [28,29].

### 3.6. Microstructure Observation

Micrographs of treated and untreated WPI solutions are shown in Figure 3. Samples (10%, *w*/*w*) treated with 0 and 15 kGy gamma radiation have shown completely different surface structures (Figure 3A). The non-irradiated protein surface has larger gaps and looser structures, while irradiated proteins are more tightly linked. These results indicate that low irradiation intensity (15 kGy) can alter the macroscopic properties (viscosity and turbidity) of proteins by changing their ultrastructures. Consistently, this conclusion seems to be tenable at high protein concentrations (Figure 3B). It seems protein–protein interaction was accelerated and enhanced by gamma radiation. In Figure 3C, the non-irradiated protein surface is smooth and shiny. However, gamma radiation causes the protein surface to form folds and bulges, and to lose the surface luster. When the irradiation dose reached 20 kGy, the protein surface structure was significantly changed. Gamma radiation has the effect on conformational change of whey protein to aggregate higher molecular weight.

### 3.7. Molecular Docking of WPI and GTA

By analyzing the interaction mode between the compound and the target protein, we can understand the mechanism of interaction between the compound and protein residues, such as hydrogen bond, π–π interaction, hydrophobic interaction, and so on and then refer to the docking score of compounds to speculate whether the screened compounds have certain activity [30,31]. The molecular docking results were obtained by the method mentioned above (Table 3). In addition, the complex of protein and small molecule was conducted by PyMOL 2.1 visual, as shown in Figure 4.

### 3.8. Lap-Shear Bonding Strength

Protein/GTA tissue adhesive was tested for bonding strength within six months after radiation. GTA (8% and 10%), used as a crosslinking agent in this study, reacted with various concentrations of WPI solutions before and after gamma radiation treatment. The workflow and the results of bonding strength are shown in Figure 5 and Figure 6, respectively.

Bonding strength was generally increased with the increasing dosage of gamma radiation and also with the increasing concentrations of whey protein solution and GTA. During storage, bonding strength showed an increase when the concentrations of protein, GTA, and dosage of gamma radiation are as constants. Generally, on the premise that the protein concentration, GTA concentration, and irradiation dose remain unchanged, bonding strength increases with the pass of time. Of course, higher irradiation doses could also accelerate the gelation of protein. Cross-linked and denatured proteins after gamma radiation can cause a lack of functional and structural properties for tissue adhesive usage.

Bonding strength is influenced by many other factors during the operation. For instance, after mixing WPI solution and GTA, the tissue adhesive may flow away from the bonding area before solidification which may decrease the bonding strength. Generally, providing the same percentage of GTA, higher dosage of gamma radiation leads to higher bonding strength. Higher concentration of protein also increases the bonding strength. Enhancement in the percentage of protein also provides a method to improve bonding strength when the concentration of GTA and the dosage of gamma radiation remain invariable. The oligomers remain stable and combine into larger aggregates at higher concentration. The higher concentration involves higher numbers of ε-NH and carbonyl groups from protein and GTA, which are the two functional compositions that agglutinated to launch the bonding strength [3].

## 4. Discussion

The revolutions of the development of the aseptic technique and noninvasive surgical techniques have been dramatically changed throughout human history [33,34]. Despite the advancements of modern technologies that have potentiated the spanking evolution of medical technique and the generation of novel surgical strategies, traditional mechanical closure techniques, such as suturing and clipping, have not modified, especially suitable for tissue anastomosis and wound repair [2]. A bio-adhesive is a material that can bond tissues together when applied on their surface, preventing separation by transferring the applied loads from one tissue to the other [4]. To avoid the pitfalls of mechanical closure techniques, the use of tissue adhesives is a potentially viable alternative. Although many polymeric compositions have been exploited as bio-adhesives, few are able to successfully bond to biological surfaces.

Recent advances in tissue adhesive concentrate on seeking more biocompatible, biodegradable, and non-toxic biomaterials with the ability to mimic the natural physiological environment and favor enhanced cell–material interactions [9]. Protein-based biomaterials have been reported with the features of enhanced bioresorbability, biocompatibility, tissue healing, and promoting regeneration [35,36]. WPI and GTA are the two key components that affect the bonding properties of the bio-adhesive. The excellent tensile and shear strength, rapid availability, and fast polymerization make protein-based sealant well-suited for cardiovascular and pulmonary repair [4]. WPI also possesses features of gelling, thickening, and water-binding capacity, resulting it to be applied in bio-adhesives [11]. Our previous study has reported that the WPI/GTA bio-adhesive system presents a comparable adhesive strength to BioGlue^®^ control [3]. However, because of the diaphragmatic paralysis, postoperative respiratory failure, and even death caused by high proportion of GTA, the concentration of GTA should be restricted in the safe range of below 10%. Therefore, the other component, WPI, has to be modified to enhance efficacy and to infinitely reduce complications. Gamma radiation is undoubtedly one of the most suitable strategies.

The compound GTA was docked with α-lactalbumin (α-LA) and β-lactoglobulin (β-LG) target proteins, respectively. The results of molecular docking showed the tight-binding of the compound with the target protein and stable matching. The complex formed by the docking compound and protein was visualized using PyMOL 2.1 software to obtain the binding mode of the compound and protein (Figure 5A). According to the binding mode, we can clearly see the combined amino acid residues of compound and protein pocket (Figure 5B). The active amino acid residues that GTA interacts with α-LA protein include Glu-49, Gln-54, Trp-104, Tyr-103, Lys-58, Asn-56, and so on. The carbonyl group of GTA can form a strong hydrogen bond to interact with amino acids (Lys-58 and Asn-56) of the proton (Figure 5C). The hydrogen bond lengths are 2.9 Å and 2.3 Å, respectively, with strong binding ability, which is of great significance for anchoring the ligand molecules in the protein pocket. In addition, the hydrophobic chain of GTA can form an affinitive and strong σ-π conjugate interaction with hydrophobic amino acid (Tyr-103), which also plays an important role in stabilizing small molecules. It can also be found from Figure 5D that the GTA compound has a high matching degree with the protein pocket and can form an intense interaction with the groove of the pocket. GTA compound forms strong hydrogen bond interaction with amino acids Lys-100, Lys-101, and Tyr-102 in β-LG protein pocket, which can effectively promote the formation of stable complexes between small molecules and proteins (Figure 5F).

Moreover, we have reported that WPI solutions and GTA were applied to the development of bio-adhesive applications which were sterilized and improved the adhesive properties of WPI by gamma radiation in this study. Gamma radiation has been used in the food industry for decades because of its low costs, improvement of the mechanical strength of protein-based bio-adhesive, and elimination of bacterial contamination [37]. While the interaction of the Co-60 gamma radiation photons via primarily photelectric, Compton scattering, and pair-production, the interactions of gamma photons with objects induce ionizations leading to the formation of ions and expelled fast-moving electrons [38]. The ions generated by radiolysis mainly undergo various chemical reactions through deprotonation reactions leading to the formation of C-centered radicals. In the presence of oxygen, low-dose irradiations, such as Co-60 gamma rays, promote the reaction through oxidation and forms the corresponding C-centered radicals [38]. Free radicals are usually produced when polymers are dry-irradiated in the presence of oxygen [39]. In aqueous solution, gamma radiation acts first on the water molecules producing active species such as hydroxyl radicals (^•^OH), superoxide anions ( O_2_^−•^), and hydrated electrons (e_aq_^−^) which in turn react with the protein molecules [18,39]. The damage to the polypeptides chains is mostly associated with the ^•^OH radicals or superoxide anions which promote the formation of C-centered radicals by abstraction of hydrogen from amino acid residues [18]. The as-obtained hydroxyl radicals or superoxide anions may react with protein as expressed in Equation (1) [40].
^•^OH/O_2_^−•^ + Protein→Protein^•^ + H_2_O_2_ or O_2_ or H_2_O(1)

Two protein radicals may combine to form a protein cross-linked structure as described in Equation (2) [40].
Protein^•^ + Protein^•^→Protein(2)

Protein• radical may also react with Protein, leading to hydroxyproline-containing peptides Equation (3) [40].
Protein^•^ + Protein→peptides + products(3)

Equations (1)–(3) represent possible mechanisms of protein degradation during gamma radiation. The formation of larger oligomers was dependent on the increase of radiation doses. The enhancement of mechanical strength in gamma-irradiated protein-based solutions triggers from the formation of cross-links which endow with elastomeric characteristics to the biomaterial [37]. The greater number of dimers in the irradiated solutions greatly increased the capability of dissociation of dimers into monomers and subsequent the efficiency of aggregation [16]. Compared with un-irradiated solutions, the irradiated coagulated solutions formed a more uniform reticular structure with better ordered structural characteristics cross-linking by β-LG and α-LA after further water loss. Therefore, that is why irradiated WPI solutions have higher mechanical strength than un-irradiated WPI solutions.

Finally, the results of this study show that the goal of developing biological tissue adhesive can be achieved through the cross-linking of a certain concentration of WPI and GTA, and the bactericidal effect on WPI and the characteristics of increasing the cross-linking strength can be accomplished by gamma irradiation. In this study, WPI was dissolved with deionized water and stored at room temperature (23 °C). The conditions, included water activity, temperature, oxygen, food, and acidity, are suitable for aerobic mesophilic bacteria to generate and grow. Aerobic count Petrifilms were incubated for 48 h at 35 °C and Yeast and Molds Petrifilms were incubated for 48–96 h at room temperature that created optimum conditions. Not surprisingly, microorganisms have not grown or generated during six-month storage. Spores, which are formed in unfavorable conditions by bacteria and are able to develop into new organisms under favorable conditions, did not grow and generate in the systems. Gamma irradiation has long been applied in the food industry as a means of ameliorating the shelf life of different food and eliminating bacterial contamination [37]. Other reports have also proved that gamma irradiation treatment could be used to improve safety conditions [3,15]. The primary mechanism by which gamma irradiation inhibits microbial growth is by breaking chemical bonds within the DNA molecule of bacteria or altering membrane permeability and other cellular functions [41].

## 5. Conclusions

To our knowledge, this study for the first time consistently demonstrates the enhancement of bonding strength and other properties of WPI solutions for six months after gamma irradiation. In this study, we revealed that the relationship of bonding strength and protein properties changed by gamma radiation and demonstrated the mechanism of protein-GTA cross-linking reaction. Our results indicated that WPI/GTA system would be a suitable alternative for traditional tissue adhesive formula. Furthermore, gamma radiation may be suitable for sterilizing the protein-based adhesive system and other dairy products. However, more in vitro and in vivo investigations are needed to evaluate for practical application.

## Figures and Tables

**Figure 1 polymers-14-03805-f001:**
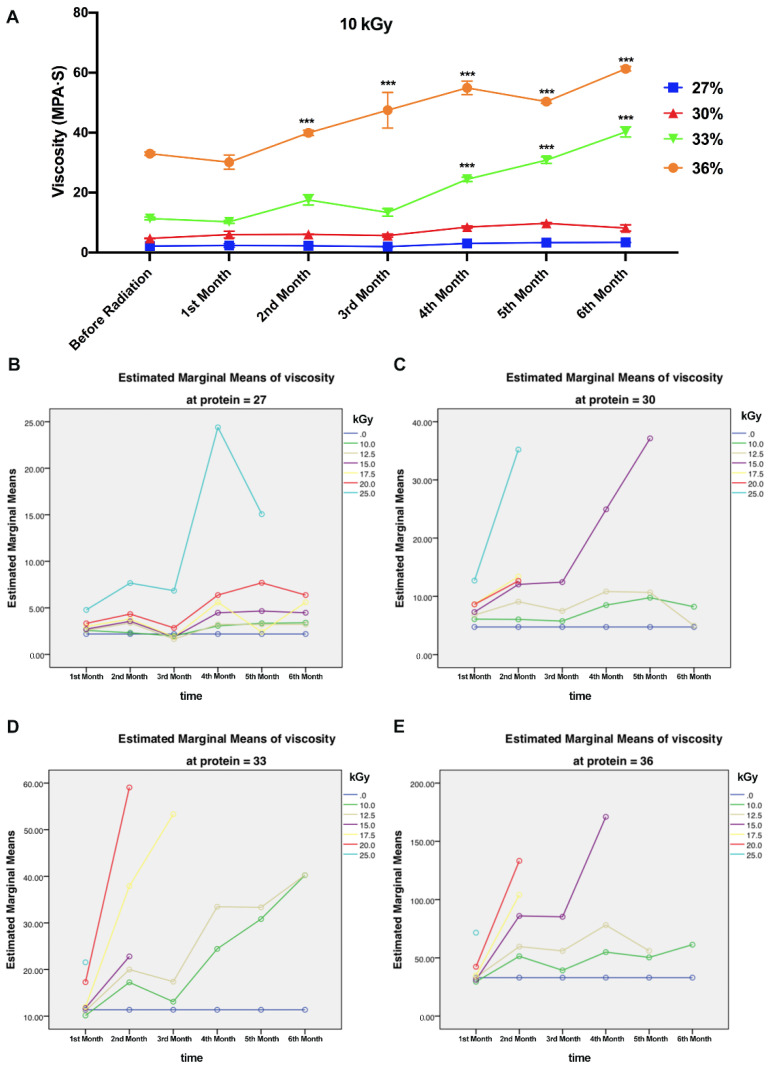
Effect of gamma radiation on viscosity of WPI solutions (10% protein). (**A**) The effects of gamma radiation (0–25 kGy) on viscosity of WPI solutions (27–36%) after gamma radiation. (**B**–**E**) Estimated marginal means of viscosity of WPI solutions, respectively, during six months. (*** *p* < 0.001).

**Figure 2 polymers-14-03805-f002:**
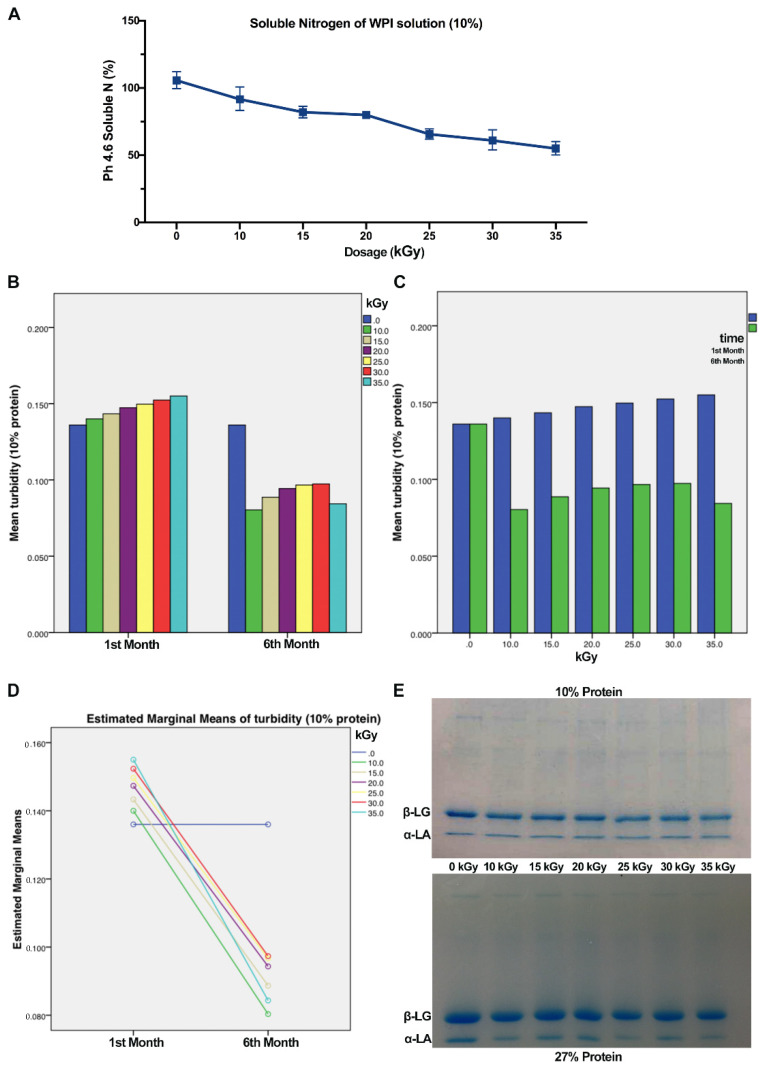
Effects of gamma radiation on property changes of WPI solutions (10% protein). (**A**) Effect of gamma radiation on soluble nitrogen of WPI solution (10%) in the first month after gamma radiation. (**B**) Comparison of estimated marginal means of turbidity of WPI solution from 0 kGy to 35 kGy gamma radiation in first month and sixth month. (**C**) Comparison of estimated marginal means of turbidity of WPI solution from first month to sixth month treated by gamma radiation (0–35 kGy). (**D**) Estimated marginal means of turbidity (10% protein) between first month and sixth month after gamma radiation 0. (**E**) Whey protein profiles before and after radiation at different dosages (10–25 kGy) of 10% and 27% WPI solutions.

**Figure 3 polymers-14-03805-f003:**
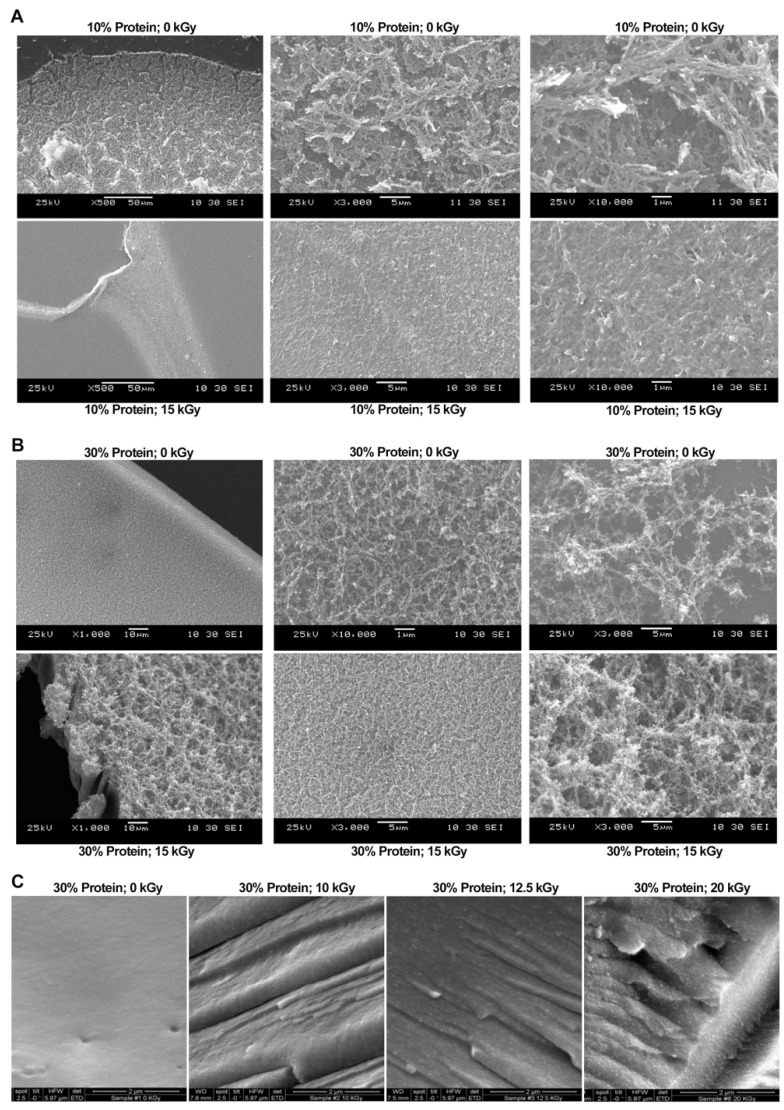
Comparison of SEM images of WPI solutions (30% protein) before and after gamma radiation treatment. (**A**) Comparison of WPI solution (10% protein) untreated or treated by 15 kGy gamma radiation. (**B**) Comparison of WPI solution (30% protein) united or treated by 15 kGy. (**C**) Comparison of WPI solution (30%) untreated or treated by 10, 12.5, 20 kGy gamma radiation.

**Figure 4 polymers-14-03805-f004:**
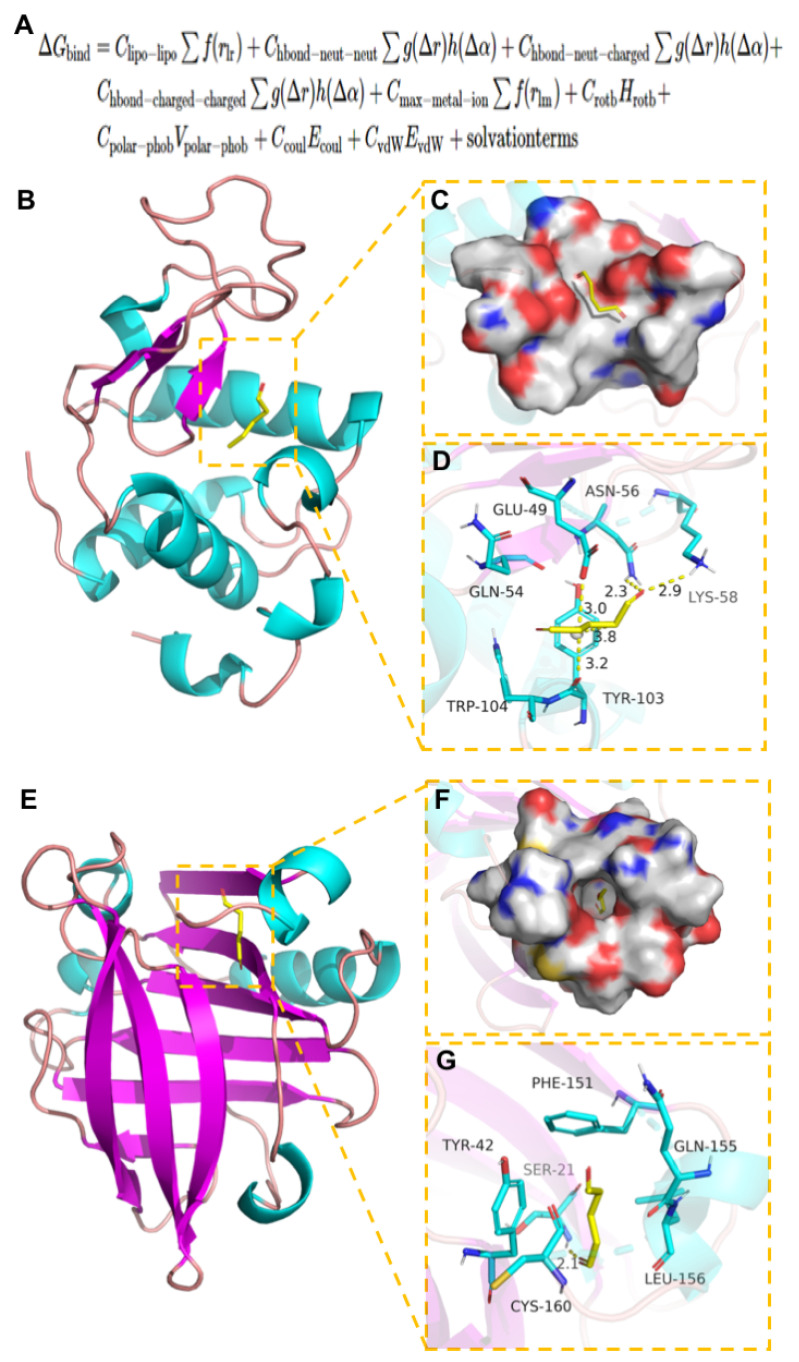
The binding mode of α-LA and β-LG proteins with GTA. (**A**) The Binding energy function. (**B**) The 3D structure of complex. (**C**) The surface of active site. (**D**) The detail binding mode of complex. GTA compound is showed in yellow. Yellow dash represents hydrogen bond distance or π-stacking. (**E**) The 3D structure of complex. (**F**) The surface of active site. (**G**) The detail binding mode of complex. GTA compound is showed in yellow. Yellow dash represents hydrogen bond distance.

**Figure 5 polymers-14-03805-f005:**
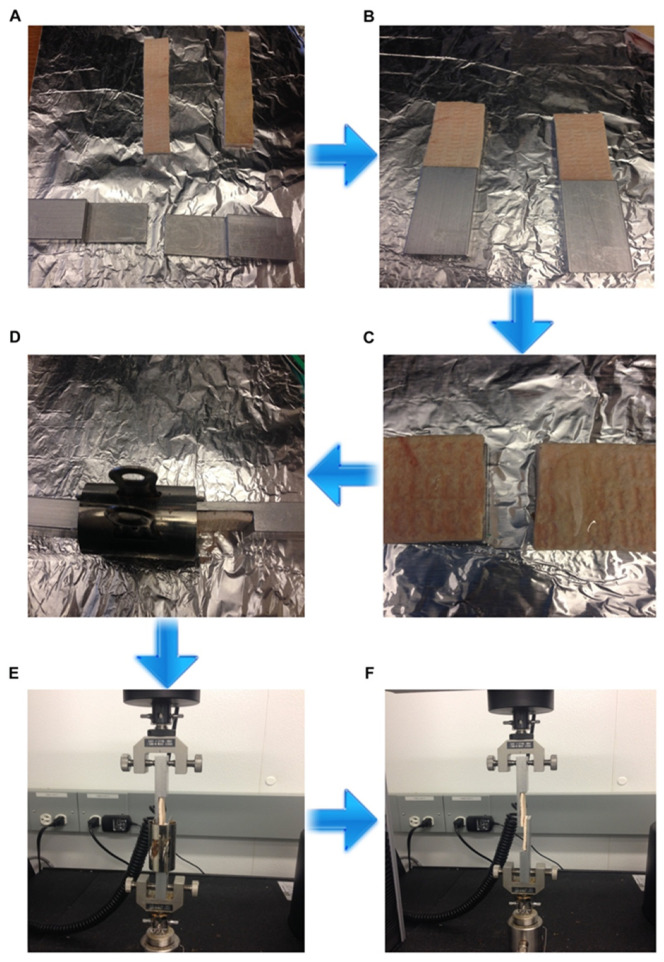
Schematic diagram of bonding strength process. (**A**) The skin graft was cut into the dimensions of 5.08 cm × 2.54 cm × 0.24 cm with a #20 Uniblade^TM^ disposable surgical scalpel (AD Surgical, Sunnyvale, CA, USA). (**B**) The skin strip was glued on an aluminum block with dermal side up by using a Loctite super glue (Henkel Corporation, Rocky Hill, CT, USA). A total of 80 μL of WPI solution and 20 μL of GTA solution were applied on the skin dermal side and mixed with a small steel spatula and then the two porcine skin strips were overlapped together by the fixture (**C**). The bonding area was 2.54 cm × 1.0 cm. Glued specimens were kept for 30 min (**D**) and tested by an Instron 5566 machine (Instron Corporation, Norwood, MA, USA). (**E**,**F**) Specimens were put tightly in the loading cell of the Instron and the test was started until the two porcine skin stripes separated. The maximum load (N) was recorded and the bonding strength (kPa) was calculated.

**Figure 6 polymers-14-03805-f006:**
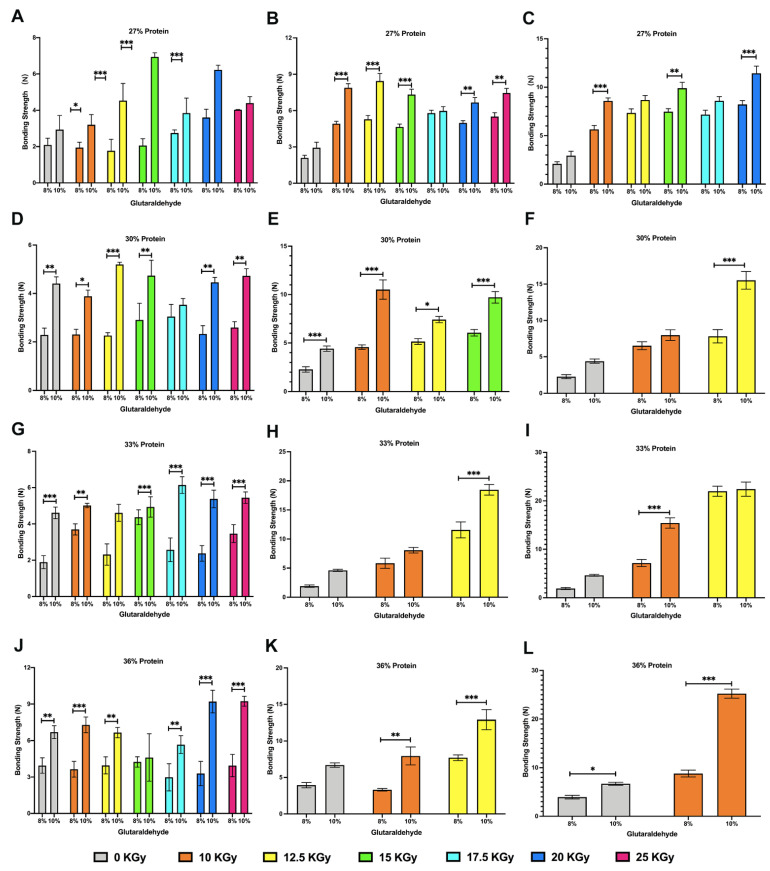
Relationship between irradiation intensity, storage time, and cross-linker and WPI solution concentrations. (**A**–**C**) The bonding strength of 27% WPI solution and GTA (8% and 10%) with different irradiation dosages after the first, third, and the fifth month of gamma radiation, respectively. 25 kGy gamma irradiated WPI formed gelation in the during period. Bonding strength cannot be conducted. (**D**–**F**) The bonding strength of 30% WPI solution and GTA (8% and 10%) with different irradiation dosages after the first, third, and the fifth month of gamma radiation, respectively. High-dosage irradiated WPI solutions formed gelation in last two periods. (**G**–**I**) The bonding strength of 30% WPI solution and GTA (8% and 10%) with different irradiation dosages after the first, third, and the fifth month of gamma radiation, respectively. 15–25 kGy irradiated WPI solutions cannot be used in bonding strength in last two periods. (**J**) The bonding strength of 36% WPI solution and GTA (8% and 10%) with different irradiation dosages after the first month of gamma radiation. (**K**) The bonding strength of 36% WPI solution and GTA (8% and 10%) with different irradiation dosages after the third month of gamma radiation. (**L**) The bonding strength of 36% WPI solution and GTA (8% and 10%) with different irradiation dosages in the fifth month of gamma radiation. Only 10 kGy irradiated WPI solutions can be used in last test (* *p* < 0.5; ** *p* < 0.1; *** *p* < 0.01).

**Table 1 polymers-14-03805-t001:** The aerobic counts and yeast and mold count of 10% WPI solutions after radiation.

	10 kGy	12.5 kGy	15 kGy	17.5 kGy	20 kGy	25 kGy
1st Month	N	N	N	N	N	N
2nd Month	N	N	N	N	N	N
3rd Month	N	N	N	N	N	N
4th Month	N	N	N	N	N	N
5th Month	N	N	N	N	N	N
6th Month	N	N	N	N	N	N

**Table 2 polymers-14-03805-t002:** The observation of sample gelation during six months after radiation.

	10 kGy	12.5 kGy	15 kGy	17.5 kGy	20 kGy	25 kGy
27% protein	N	N	N	N	N	Y
30% protein	N	N	Y	Y	Y	Y
33% protein	N	N	Y	Y	Y	Y
36% protein	N	Y	Y	Y	Y	Y

**Table 3 polymers-14-03805-t003:** Target protein docking results.

Target	Compound	CompoundStructure	Binding Energy(kJ/mol)
α-LA	GTA	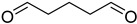	−5.75
β-LG	−4.93

Note: Binding energy function is shown in Figure 4A [32].

## Data Availability

Not applicable.

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
