# Peer review of "Effects of Radiation on Cross-Linking Reaction, Microstructure, and Microbiological Properties of Whey Protein-Based Tissue Adhesive Development"

_polymers, 2022, doi:10.3390/polym14183805_

Round 1

Reviewer 1 Report

In this manuscript, the author studied the effect of radiation on whey protein in the aspects of protein-protein interaction, microstructure, and microbiological properties for the application in tissue adhesive. They found gamma radiation and storage time can significantly increase the viscosity of whey protein solutions. The extent of oligomerization of whey protein isolate solutions are increased by the enhancement in gamma radiation intensity. They concluded the combination of gamma irradiated whey protein isolate solutions and glutaraldehyde can be used as a novel biomaterial tissue adhesive. Overall, it is an interesting study, and the results are sound. However, some of the data were not well presented. I recommend publication of this manuscript in polymers after the following major issues are addressed.

1) Some of mistakes in the text should be corrected, such as “10%, 27, 30%, 33%,”

2) The fonts in the figures should be revised to be consistent in size and clearly visible. The quality of the images should be improved to meet scientific quality.

3) The authors should explain why the viscosity of Whey protein isolate with low content (27% and 30%) did not change under gamma irradiation in Figure 1A.

4) The effect of gamma radiation on the viscosity of WPI was tracked up to 6 months. The authors should give a more detailed explanation on the experimental design. Some of the details should be provided with the figure caption, such as the length of irradiation.

5) Figure 2E is not explained in the text. The result of gel electrophoresis should be analyzed and discussed.

6) In Figure 6, the authors should demonstrate the process more clearly and a description should be added in the caption.

7) In Figure 7, the authors should reorganize the images to avoid space.

Author Response

Responses to Reviewer#1:

1) Some of mistakes in the text should be corrected, such as “10%, 27, 30%, 33%,”

Answer: Thank you for your valuable concern. We have checked for mistakes and revised them, as suggested.

2) The fonts in the figures should be revised to be consistent in size and clearly visible. The quality of the images should be improved to meet scientific quality.

Answer: Thank you for your valuable concern. The fonts in figures were revised to be consistent in size and clearly visible. The quality of the images was also improved to meet scientific quality.

3) The authors should explain why the viscosity of Whey protein isolate with low content (27% and 30%) did not change under gamma irradiation in Figure 1A.

Answer: Thank you for your valuable concern. We have explained why the viscosity of Whey protein isolate with low content did not change under gamma irradiation in Lines 245-247.

4) The effect of gamma radiation on the viscosity of WPI was tracked up to 6 months. The authors should give a more detailed explanation on the experimental design. Some of the details should be provided with the figure caption, such as the length of irradiation.

Answer: Thank you for your valuable concern. We have added details of gamma radiation in Lines 112-113.

5) Figure 2E is not explained in the text. The result of gel electrophoresis should be analyzed and discussed.

Answer: Thank you for your valuable concern. The result of gel electrophoresis was added in Lines 314-328.

6) In Figure 6, the authors should demonstrate the process more clearly and a description should be added in the caption.

Answer: Thank you for your valuable concern. Figure 6 was corrected to Figure 5 and explained in the figure caption in Lines 444-453.

7) In Figure 7, the authors should reorganize the images to avoid space.

Answer: Thank you for your valuable concern. Figure 7 was corrected to Figure 6 and reorganized, as suggested.

Reviewer 2 Report

Dear Authors

The presented work is very interesting and the experimental design is satisfactory. I can recommend the publication of your work in its current form.

The authors studied for the first time the using of developed radiated crosslinked whey protein isolate as a novel biomaterial tissue adhesive.

The topic of the manuscript is original and the research plan of the manuscript is well designed. The authors merged the benefits of both radiation crosslinking, using Gamma radiation, and the chemical crosslinking using Glutaraldehyde.

The authors characterized the developed WPI adhesive very well using the proper techniques and correlate the preparation conditions to the obtained characters reaching to the optimum conditions. The manuscript is well written and easy to follow with interesting. Finally, the conclusion of the manuscript is supports with the obtained results and opens a new area of interest research topic for an old challenge in the field of developing wound healling materials.  

Accordingly, I highly recommend the publication of the manuscript in its current form.

Author Response

Thank you for your high recognition and support

Round 2

Reviewer 1 Report

The authors have addressed the comments. Although the quality of the figures remains poor, the overall quality of the manuscript has been improved. The authors may further improve the image quality in the production process. Thus, I recommend the acceptance of the manuscript.